# Clustering Analysis Identified Distinct Clinical Phenotypes among Hemodialysis Patients in Their Immunological Response to the BNT162b2 mRNA Vaccine against SARS-CoV-2

**DOI:** 10.3390/vaccines12101150

**Published:** 2024-10-08

**Authors:** Guy Rostoker, Stéphanie Rouanet, Mireille Griuncelli, Christelle Loridon, Ghada Boulahia, Luc Gagnon

**Affiliations:** 1Department of Nephrology and Dialysis, Claude Galien Private Hospital—Ramsay Health Care, 91480 Quincy-sous-Sénart, France; 2Collège de Médecine des Hôpitaux de Paris, 75610 Paris, France; 3StatEthic, 92300 Levallois-Perret, France; 4IQVIA Laboratories Vaccines, Laval, QC H7V 3S8, Canada

**Keywords:** clustering, dialysis, IgG anti-spike antibodies, mRNA vaccination, SARS-CoV-2

## Abstract

**Background:** The 2019 coronavirus disease (COVID-19) pandemic induced a major health crisis worldwide, notably among end-stage kidney disease (ESKD) patients. Vaccination against SARS-CoV-2, especially with messenger RNA (mRNA) vaccines, is highly effective and reduces hospitalization and mortality in both the general and ESKD populations. Age and previous COVID-19 infection have been identified as major determinants of the vaccine response in both the general population and ESKD patients. **Methods:** To determine the specific phenotype of ESKD patients in relation to their vaccine response, a clustering approach was used in a cohort of 117 fully vaccinated patients. **Results:** Clustering revealed three distinct clinical phenotypes among hemodialysis patients in terms of immunological response. Two clusters, consisting of either women with a long dialysis history or male subjects with diabetes with a moderate history of dialysis, exhibited low levels of IgG anti-spike antibodies. The third cluster consisted of non-diabetic middle-aged men with a moderate dialysis vintage and a very good serological response to vaccination. **Conclusions:** These vaccinal phenotypes of dialysis patients are easily identifiable in current practice, allowing for differential serological follow-up and tailored booster SARS-CoV-2 vaccination.

## 1. Introduction

The 2019 coronavirus disease (COVID-19) pandemic induced a major health crisis worldwide, with high vulnerability seen among end-stage kidney disease (ESKD) patients. They experienced high mortality rates, notably during the first wave [1]. Vaccination against SARS-CoV-2, especially with messenger RNA (mRNA) vaccines, has been shown to be highly effective and to reduce hospitalization and mortality rates in both the general population and ESKD patients [2,3]. Age and previous COVID-19 infection have been shown to be major determinants of the vaccine response in both the general population and in ESKD patients [1,2,3]. This is best appreciated by neutralizing antibodies and, in clinical practice, by anti-spike antibodies, which are considered to be surrogate markers [2,4]. In addition to age and previous COVID-19 infection, multivariable analyses have shown that malnutrition, comorbidities, and previous immunosuppressive treatment can negatively impact the vaccine response in ESKD patients [5,6,7].

Of note, evaluating SARS-CoV-2 vaccination in ESKD has many similarities with the evaluation of patients in some other medical conditions, notably inflammatory bowel disease treated by biological or anti-metabolite drugs where patients are also immunocompromised [8,9].

Cluster analysis, also known as clustering or numerical taxonomy, is a statistical method used to group objects or subjects based on their similarities. The method aims to create clusters of objects or subjects that share common characteristics and differ from those in other clusters [10]. This technique is widely used for statistical data analysis in various fields, including pattern recognition, image analysis, bioinformatics, data compression, computer graphics, machine learning, psychology, sociology, economics, and medicine [10]. Surprisingly, no cluster analysis in patients on dialysis focusing on the analysis of vaccine response against COVID is available to date.

Our objective was therefore to conduct a cluster analysis of specific phenotypes in ESKD patients based on clinical and dialysis characteristics and on their response to full vaccination against SARS-CoV-2 with BNT162b2 mRNA vaccine. Our hypothesis was that providing nephrologists with differential serological follow-up of IgG anti-spike antibodies and tailored booster vaccination of their ESKD patients on dialysis could be beneficial.

## 2. Materials and Methods

### 2.1. Patients and Study Design

This retrospective serological study analyzed the quality and determinants of the vaccine response within one to three months following a full vaccination with two doses of the Pfizer original BNT162b2 mRNA vaccine. This study was conducted from January 2021 to December 2021, as soon as COVID-19 vaccines became available in France, on a cohort of hemodialysis patients treated at Claude Galien Hospital dialysis center.

#### 2.1.1. Study Population

This study included 117 patients treated in the dialysis center of Claude Galien Private Hospital (Quincy-sous-Sénart, Great Paris area, France). Patients had 3 dialysis sessions per week, by classical hemodialysis, hemodiafiltration or expanded hemodialysis (HDx).

The inclusion criteria were as follows:−Male or female patients, over 18 years of age.−Patients undergoing hemodialysis at the dialysis center of Claude Galien Hospital.−Patients vaccinated for the first time against COVID-19.−Patients informed and agreeing to participate to the study.−Patients affiliated to or benefiting from a social security scheme.−The non-inclusion criteria were as follows:−Patients contraindicated to vaccination or not vaccinated.−Administration of any other vaccine within the previous 3 weeks.−Protected patients: adults under guardianship, curatorship, or other legal protection, deprived of liberty by judicial or administrative decision.−Pregnant women.

#### 2.1.2. Study Design

We conducted a monocentric observational study with a retrospective analysis of health data usually collected as part of routine care, with longitudinal follow-up of vaccinated patients for one year.

The collected data were anti-SARS-CoV-2 antibody levels; demographic characteristics including age (years) and sex, dialysis vintage (months); type of dialysis: traditional hemodialysis on polysulphone membrane, hemodiafiltration, hemodialysis on adsorbent membrane (PMMA, AN69), expanded hemodialysis (HDx); presence of diabetes mellitus; Charlson index modified according to age; biomarkers performed routinely for inflammation and nutrition status; treatment of anemia: quantity of IV iron (expressed as mg/month) received in the year preceding the first vaccination dose as the quantity of erythropoietin (ARANESP/Darbepoetin) received (expressed as μg/week) in the year preceding the first vaccination dose.

The main aim of this research was to describe the humoral immune response of dialysis patients one to three months after full vaccination against COVID-19 (with the Pfizer BNT162b2 mRNA vaccine).

One of the secondary aims of this research (studied here) was to determine patient profiles based on anti-SARS-CoV-2 IgG levels, demographic characteristics, dialysis modalities, comorbidities, inflammation, nutrition status, and anemia treatment with ESA and IV iron therapy.

### 2.2. Determination of IgG Anti-SARS-CoV-2 Asntibodies

IgG anti-SARS-CoV-2 spike antibodies were determined as part of the routine biological follow-up of patients (conducted monthly during the three first months after the second vaccine dose and quarterly afterwards) by Cerballiance laboratory (Ile-de-France Sud, Lisses, France) using Chemiluminescence ALINITY from Abbott. The results were expressed in UA/mL and later transposed to BAU/mL for this study using the equation BAU/mL = 1/7 UA/mL.

### 2.3. Analysis of Dialysis Charts and Medical Records

A clinical research technician (MG) carefully reviewed the dialysis charts and biological and medical records, which were then checked by another clinical research technician (CL).

### 2.4. Statistical Analyses

The statistical analyses were performed on complete cases (all patients with no missing data). To determine the specific phenotype of ESKD patients, a clustering approach was used. This statistical method involves grouping a set of observations based on similarities, resulting in distinct clusters that differ from one another.

Clusters were identified based on the following variables: anti-Spike IgG level after the second vaccine dose, age, sex, dialysis vintage, type of hemodialysis technique, diabetes mellitus, modified Charlson’s comorbidity index, C-reactive protein (CRP), albumin, pre-albumin, amount of erythropoiesis-stimulating agents (ESAs) given in the year prior to vaccination, and amount of intravenous (IV) iron given in the year prior to vaccination. The anti-Spike IgG, dialysis vintage, and CRP variables were log-transformed to conform more closely to a normal distribution.

The clustering was performed using the PAM (partitioning around medoids) method with Gower’s distance [10,11]. Fifty random initializations were performed, and the Silhouette statistic was used to determine the optimal number of clusters [12].

The results for continuous variables were reported as medians (IQR) for the obtained clusters and compared using the Kruskal–Wallis test. The results for categorical variables were reported as numbers and percentages for the obtained clusters and compared using Fisher’s exact test. Statistical analyses were performed using R Statistical Software version 4.3.1. For all statistical tests, *p*-values less than 0.05 were considered statistically significant.

## 3. Results

This study included 117 hemodialysis patients. The main characteristics of this cohort are shown in Table 1. Briefly, this cohort comprised around 60% of men and 43% of subjects with diabetes. Median age was 68 years. The median dialysis vintage was 41 months and the median modified Charlson index was 7. This cohort has many similarities with the overall population of dialysis patients in France and in western Europe (Table 1).

The patients were divided into three clusters based on the proposed variables. Cluster 1 and 2 each had 40 patients, while Cluster 3 had 37 patients. The main characteristics of these three clusters are shown in Table 2 and Figure 1 and Figure 2. Cluster 1 was mainly composed of relatively young men (90%) without diabetes, with a moderate dialysis vintage, a low modified Charlson’s comorbidity index, and a strong serological response to vaccination. In contrast, Cluster 3 was composed of older women (84%), mainly non-diabetic, with a long dialysis vintage, a low modified Charlson’s comorbidity index, and a poor vaccine response. Lastly, Cluster 2 was characterized by a poor vaccine response in a group of mainly male diabetic patients with a high modified Charlson’s comorbidity index, higher CRP, and moderate dialysis vintage (Table 2 and Figure 1 and Figure 2).

Finally, caution should be exercised when analyzing differences in hemodialysis techniques. It is important to note that most patients on hemodiafiltration and expanded hemodialysis (HDx) were in Cluster 3, which is composed of women with a long dialysis vintage. This suggests that these techniques may be more suitable for patients who have been on the transplantation waiting list for a longer period, mainly due to sensitization caused by multiple pregnancies.

## 4. Discussion

To our knowledge, this is the first cluster analysis devoted to the vaccine response against COVID-19 in dialysis patients. Interestingly, this clustering analysis revealed three distinct clinical phenotypes among hemodialysis patients, in relation with their immunological response to the BNT162b2 mRNA vaccine. Two of the clusters comprised individuals with low IgG anti-spike antibodies. The first one comprised women with a long dialysis vintage, and the other cluster was composed of male diabetic patients with a moderate dialysis vintage. The third cluster was composed of non-diabetic middle-aged men with a moderate dialysis vintage and a strong serological response to vaccination.

Since dialysis patients have a reduced immune response to SARS-CoV-2 vaccination with lower antibody levels and faster waning, monitoring of anti-spike antibodies as a correlate of protection to guide future booster vaccines has been universally, regularly, and indiscriminately applicated since 2021 in the dialysis population [4,13]. Here, we show that a precision medicine strategy for SARS-CoV-2 vaccination follow-up may be adopted in ESKD patients based on easily identified clinical phenotypes defined by dialysis vintage, diabetic status, and gender, subject to similar findings being observed in a mandatory validation cohort.

The poor vaccine response identified in this study was mainly related first to diabetic status per se, which has been shown to be associated with an impaired immune response to SARS-CoV-2 vaccine seen even in patients with normal renal function [14,15] and secondly to the time spent on dialysis, which linearly worsens secondary immunodeficiency related to kidney disease (SIDKD).

SIDKD is characterized by defective innate immunity, a reduced antigen-presenting capability of dendritic cells and macrophages, and impaired B and T cell function. The major factors of these anomalies are uremic enteropathy and dysbiosis leading to increased bacterial and endotoxin translocation, favoring systemic inflammation together with an accumulation of immunoregulatory metabolites and proteins related to impaired urinary clearance and the limitation of actual dialyzers to clear uremic toxins of high molecular weight and those linked to serum albumin [16].

Given the poorer vaccine response in two clusters of patients (namely Cluster 2 and Cluster 3) and evidence suggesting a higher efficacy of the mRNA-1273 over the BNT162b2 mRNA vaccine, which has been consistently shown in ESKD patients, it is tempting to assume that, firstly, specific studies on Moderna’s Spikevax (mRNA-1273) aimed at overcoming this vaccine hypo-response in the subpopulation of patients on dialysis are highly desirable and, secondly, that in clinical practice, patients carrying the phenotypes of Clusters 2 and 3 should be prioritized for a booster dose with mRNA-1273 instead of the BNT162b2 mRNA vaccine [17,18,19].

The main limitation of our study relates to its design, as all the patients were recruited at a single dialysis center at a French hospital, using only the BNT162b2 mRNA vaccine, which could limit the generalization of the findings. Of note, cluster analysis is a descriptive type of analysis which aims to identify a group structure in a dataset (i.e., among a specific population with a defined set of variables) [10,11,12]. Our study was only aimed at describing different patient profiles among ESKD patients after vaccination against COVID-19. Therefore, potential results in non-ESKD patients cannot be inferred from such an analysis. Finally, a larger panel of patients may have enabled us to identify subsets with a higher accuracy, showing sub-clusters with a higher intra-cluster homogeneity.

## 5. Conclusions

The three phenotypes characterized by cluster analysis may be easily identified in current clinical practice, allowing for differential serological follow-up and tailored booster SARS-CoV-2 vaccination among patients undergoing dialysis. This new clinical classification after mandatory validation in an external cohort may be universal and easy to perform, allowing tailored management by nephrologists and dialysis nurses for the prevention of severe SARS-CoV-2 infection in a dialysis setting.

## Figures and Tables

**Figure 1 vaccines-12-01150-f001:**
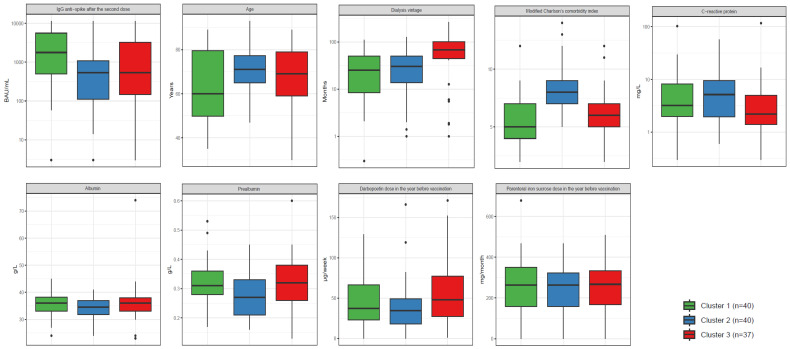
Distribution of continuous variables by cluster. The box plot indicates the minimum and maximum values at the end of the “whiskers”, the 25th and 75th percentiles at the ends of a box, and the median as a horizontal line in the box at the 50th percentile value. The remaining values of the distribution are independently represented by the black points, so that outliers can be identified.

**Figure 2 vaccines-12-01150-f002:**
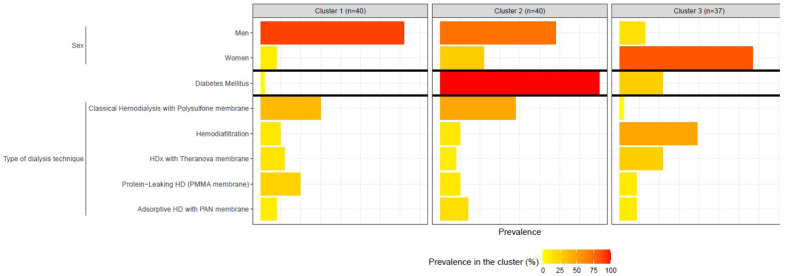
Description of discrete variables by cluster. The band highlighted in bold indicates the greatest difference in prevalence between the 3 clusters.

**Table 1 vaccines-12-01150-t001:** Characteristics of the overall cohort.

	Total n = 117
IgG anti-spike after the second dose (BAU/mL)	690.9 (149.8–3755.4)
Age (years)	68.0 (57.0–78.0)
Sex Men Women	71 (60.7%) 46 (39.3%)
Dialysis vintage (months)	41.0 (14.0–73.3)
Minimum–maximal	0.3–265.1
Type of dialysis technique (HD1) Classical HD with polysulfone membrane (HD2) Hemodiafiltration (HD3) HDx with theranova membrane (HD4) Protein-Leaking HD (PMMA membrane) (HD5) Adsorptive HD with PAN membrane	35 (29.9%) 28 (23.9%) 20 (17.1%) 19 (16.2%) 15 (12.8%)
Diabetes Mellitus Yes No	51 (43.6%) 66 (56.4%)
Modified Charlson’s comorbidity index	7 (5–8)
C-reactive protein (mg/L)	3.2 (1.8–8.2)
Albumin (g/L)	36.0 (33.0–38.0)
Prealbumin (g/L)	0.30 (0.26–0.37)
Darbepoetin dose in the yearbefore vaccination (µg/week)	36.0 (23.0–67.0)
Parenteral iron sucrose dose in the year before vaccination (mg/month)	266.0 (158.0–333.0)

**Table 2 vaccines-12-01150-t002:** Characteristics of the patients in the three clusters.

	Cluster 1 n = 40	Cluster 2 n = 40	Cluster 3 n = 37	*p*-Value *
IgG anti-spike after the second dose (BAU/mL)	1772.6 (503.5–5586.9)	537.1 (111.6–1081.2)	538.3 (145.2–3259.8)	0.0090
Age (years)	60.0 (49.8–79.5)	71.0 (65.0–77.3)	69.0 (59.0–79.0)	0.0560
Sex Men Women	36 (90.0%) 4 (10.0%)	29 (72.5%) 11 (27.5%)	6 (16.2%) 31 (83.8%)	<0.0001
Dialysis vintage (months)	25.4 (8.5–50.6)	30.4 (13.8–50.6)	68.2 (44.6–102.0)	0.0003
Minimum–maximal	0.3–110.2	1–127.8	1–265.1	
Type of dialysis technique (HD1) Classical HD with polysulfone membrane (HD2) Hemodiafiltration (HD3) HDx with theranova membrane (HD4) Protein-Leaking HD (PMMA membrane) (HD5) Adsorptive HD with PAN membrane	15 (37.5%) 5 (12.5%) 6 (15.0%) 10 (25.0%) 4 (10.0%)	19 (47.5%) 5 (12.5%) 4 (10.0%) 5 (12.5%) 7 (17.5%)	1 (2.7%) 18 (48.6%) 10 (27.0%) 4 (10.8%) 4 (10.8%)	<0.0001
Diabetes Mellitus Yes No	1 (2.5%) 39 (97.5%)	40 (100%) 0 (0.0%)	10 (27.0%) 27 (73%)	<0.0001
Modified Charlson’s comorbidity index	5 (4–7)	8 (7–9)	6 (5–7)	<0.0001
C-reactive protein (mg/L)	3.2 (2.0–8.1)	5.2 (2.0–9.5)	2.2 (1.4–5.0)	0.0520
Albumin (g/L)	36.0 (33.0–38.3)	34.5 (31.8–37.0)	36.0 (33.0–38.0)	0.2600
Prealbumin (g/L)	0.31 (0.28–0.36)	0.27 (0.21–0.33)	0.32 (0.26–0.38)	0.0390
Darbepoetin dose in the year before vaccination (µg/week)	37.0 (22.8–66.3)	34.5 (18.0–49.3)	48.0 (27.0–77.0)	0.1100
Parenteral iron sucrose dose in the year before vaccination (mg/month)	263.5 (158.5–350.0)	262.5 (158.0–323.3)	267.0 (167.0–333.0)	0.9800

Ig: immunoglobulin; BAU: binding antibody unit; HDx: expanded hemodialysis; HD: hemodialysis; PMMA: polymethyl methacrylate; PAN: polyacrylonitrile. Values are given in median (first and third quartile (Q1–Q3)) and n, number of patient (percentage of patients %). * *p* value determined using either Kruskal–Wallis or Fisher’ exact test.

## Data Availability

The deidentified and anonymized data are available upon reasonable request. Requests should be directed to rostotom@orange.fr.

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
