# Peer review of "Clustering Analysis Identified Distinct Clinical Phenotypes among Hemodialysis Patients in Their Immunological Response to the BNT162b2 mRNA Vaccine against SARS-CoV-2"

_vaccines, 2024, doi:10.3390/vaccines12101150_

Round 1

Reviewer 1 Report

Comments and Suggestions for Authors

 In the study  Clustering analysis identified distinct clinical phenotypes  among hemodialysis patients in their immunological response  to the mRNA vaccine COMIRNATY against SARS-CoV-2”,   there were  presented the  results  of the vaccine response against COVID 19 in 117 dialysis patients divided in three  clusters.  One cluster  is composed of women with a long dialysis vintage,the other cluster is composed of male diabetics with a moderate dialysis  vintage,  the third cluster is composed of non-diabetic middle-aged men with a moderate dialysis vintage and a strong serological response to vaccination. The poor vaccine response  was identified in cluster of mainly male diabetic patients with a high modified Charlson comorbidity index, higher CRP, and moderate dialysis. 

 The results of this study are important for  administration  of SARS-CoV-2 vaccination among patients undergoing dialysis. 

Reviewer 2 Report

Comments and Suggestions for Authors

Brief summary :

The authors review the immunological responses to the mRNA COMIRNATY vaccine against SARS-CoV-2 in individuals with end-stage kidney disease (ESKD). It has been established that mRNA vaccines are highly effective both in the general population and in ESKD patients. The vaccine response is influenced by several factors, including age and prior SARS-CoV-2 infection. Specifically, in ESKD patients, various types of immune responses are observed. By analyzing 117 fully vaccinated ESKD patients, the authors identified several phenotypes, each associated with distinct serological profiles and recommended timing for booster vaccinations.

Strengths:

The reviewed work not only examines the timing for the booster vaccine but also defines three main serological clusters, which can be easily identified in current clinical practice, thus providing a practical approach.

For the dataset, the PAM (Partitioning Around Medoids) method was used. This is a good choice, as the PAM method is not very sensitive to outliers. The use of Gower distances is also justified, given that each data point is represented by a mixture of categorical and numeric data.

Weaknesses:

11.      In a recent study by Belgian scientists (Debelle, F., Nguyen, V.T.P., Boitquin, L., GuillenAnaya, M.A., Gankam, F., Declèves, A.E. and CoviDial study group, 2024, March. Monitoring strategy of COVID19 vaccination in dialysis patients based on a multiplex immunodot method: The CoviDial study. In Seminars in Dialysis (Vol. 37, No. 2, pp. 145-152)), the researchers specifically reviewed Moderna’s Spikevax. In the reviewed study, however, only COMIRNATY by Pfizer was examined. Although French regulations do not recommend Moderna’s vaccine for initial vaccination, comparing the two vaccines could provide a better understanding of their differences and benefits.

22.      The number of patients differs in the “abstract” and “materials and methods” sections Are there 117 patients or 128?

33.      The study is limited by being conducted at a single center with a relatively small number of patients, which raises the question of whether these data can be generalized to the broader population.

44.      There is no control group, so it is unclear whether the data from non-ESKD patients with similar disorders would cluster in the same manner.

55.      There is no evidence on how racial, age, and sex differences were accounted for and normalized in the dataset.

66.      Since the clusters indicate that the patient population lacks certain groups, the absence of these groups in the dataset, along with the lack of diversity in patient groups, makes it impossible to validate the clustering. The full dataset is also not provided.

Reviewer 3 Report

Comments and Suggestions for Authors

The authors have conducted a clustering analysis to profile the effects of the BNT162b2 vaccine in approximately one hundred haemodialysis patients, with the aim of identifying specific clinical variables associated with the vaccine response.

The clusters identified by the authors included specific comorbidities, chronological age, and duration of dialysis.

A couple of minor considerations:

- I am not particularly fond of the use of the commercial name of the vaccine in the work (especially in the title). I would prefer definitions based on the active principle, such as "BNT162b2 mRNA vaccine".

- I would like to see, in the introduction, a comparative discussion with other conditions in which the vaccine has been thoroughly evaluated during COVID-19, for instance, in patients treated with biological drugs during COVID-19 with inflammatory bowel disease, which has been the subject of considerable debate. I recommend discussing a study that has utilised the same vaccine as yours (https://pubmed.ncbi.nlm.nih.gov/36047032/). As is well known, also haemodialysis patients can be immunocompromised.

With these minor modifications, I believe the work could be slightly improved.
